

# Potential of the economic valuation of soil-based ecosystem services to inform sustainable soil management and policy

Bartosz Bartkowski[1], Stephan Bartke[1], Katharina Helming[2,3], Carsten Paul[2], Anja-Kristina Techen[2] and Bernd Hansjürgens[1]

[1] UFZ—Helmholtz Centre for Environmental Research, Leipzig, Germany
[2] ZALF—Leibniz Centre for Agricultural Landscape Research, Müncheberg, Germany
[3] HNEE—University for Sustainable Development, Eberswalde, Germany

## ABSTRACT

The concept of ecosystem services, especially in combination with economic valuation, can illuminate trade-offs involved in soil management, policy and governance, and thus support decision making. In this paper, we investigate and highlight the potential and limitations of the economic valuation of soil-based ecosystem services to inform sustainable soil management and policy. We formulate a definition of soil-based ecosystem services as basis for conducting a review of existing soil valuation studies with a focus on the inclusion of ecosystem services and the choice of valuation methods. We find that, so far, the economic valuation of soil-based ecosystem services has covered only a small number of such services and most studies have employed cost-based methods rather than state-of-the-art preference-based valuation methods, even though the latter would better acknowledge the public good character of soil related services. Therefore, the relevance of existing valuation studies for political processes is low. Broadening the spectrum of analyzed ecosystem services as well as using preference-based methods would likely increase the informational quality and policy relevance of valuation results. We point out options for improvement based on recent advances in economic valuation theory and practice. We conclude by investigating the specific roles economic valuation results can play in different phases of the policy-making process, and the specific requirements for its usefulness in this context.

# INTRODUCTION

In recent years, scientific and political initiatives have been established to emphasize the importance of intact and well-functioning ecosystems for human well-being (*Kumar, 2010*; *Pascual et al., 2017*; *Díaz et al., 2018*). However, the attention given to the importance of soils in this context is still limited (*Dominati, Patterson & Mackay, 2010*; *Baveye, Baveye & Gowdy, 2016*). Recently, the older concept of soil functions has been complemented by applications of the ecosystem service perspective to soils (*Dominati, Patterson & Mackay, 2010*; *Adhikari & Hartemink, 2016*; *Helming et al., 2018*; *Van der Meulen et al., 2018*). This helps to illustrate the soils' societal relevance; however, research in this area is still nascent

Corresponding author
Bartosz Bartkowski,
bartosz.bartkowski@ufz.de

and needs further development. Furthermore, the use and management of soils, particularly in the agricultural context, involves trade-offs between the effects on different ecosystem services as well as biodiversity (*Schwilch et al., 2018*). There is a need to express these trade-offs in ways that help navigate them and are relevant for policy making.

One way to do this is by means of economic valuation. Economic values can be used for different policy-relevant purposes: *informative* (e.g., communication of values), *decisive* (e.g., cost–benefit analysis) and *technical* (e.g., informing the setting of agri-environmental payment levels) (*Laurans et al., 2013*). A proper understanding of the potential and limitations of economic valuation is necessary if it is to inform political decision-making processes.

In this paper, we conduct a literature review to investigate the current status and the potential of the economic valuation of soil-based ecosystem services to inform sustainable soil management and policy. We seek to contribute to the discussions of whether and to what extent economic perspectives can support the recognition and valuation of soil-based ecosystem services. The economic valuation of ecosystem services, including soil-based ones, is a controversial issue (*Baveye, Baveye & Gowdy, 2016*); whether valuation studies are conducted in the first place and how they are designed and their results interpreted has consequences for the incorporation of economic values into decision making (*Kumar, 2010*). Our main aim is to shed light on this by evaluating existing attempts to provide economic values of soil-based ecosystem services, and to use insights from economic theory and from the practice of economic valuation in order to suggest ways for improvement. Our paper should be of particular interest for non-economists working in soil-related environmental sciences.

Previous literature reviews and conceptual contributions usually focused on soil-based ecosystem services and treated economic valuation in a less systematic way (e.g., *Baveye, Baveye & Gowdy, 2016*; *Dominati, Patterson & Mackay, 2010*). An exception is *Jónsson & Davíðsdóttir*'s (*2016*) review, which focuses on the economic valuation of soils. However, their main interest is in the ranges of monetary values that can be extracted from the literature, while our primary aim is to systematically illuminate how economic valuation is used (from a methodological point of view) and which soil-based ecosystem services have been addressed in economic valuation studies to date.

We structure our paper as follows: first, we briefly introduce the methodology used in conducting the literature review ("Survey methodology"). Second, we develop an operationalized definition of soil-based ecosystem services that will guide our further analysis ("Soil-based ecosystem services"). Third, we offer a comprehensive review of existing soil valuation studies with a focus on the ecosystem services covered as well as the valuation methods applied ("Economic valuation of soil-based ecosystem services"). On this basis, we provide a critical discussion of the potential and limitations of economic valuation to inform sustainable soil management ("Economic valuation for sustainable soil management and policy"). In "Relevance of the economic valuation of soil-based ecosystem services for policy making" we conclude by deriving some more general implications for policy.

## SURVEY METHODOLOGY

The core of the present paper is a focused review of existing soil valuation studies. There is a small number of reviews of such studies, including *Jónsson & Davíðsdóttir (2016)*, *Robinson et al. (2014)*, *Baveye, Baveye & Gowdy (2016)* and the recent Mapping and Assessment of Ecosystem Services (MAES) report on Soil Ecosystems (*Van der Meulen et al., 2018*), which have different levels of comprehensiveness, different purposes and foci. To provide a comprehensive overview of the relevant literature, we use peer-reviewed studies mentioned in *Jónsson & Davíðsdóttir (2016)*, complemented by additional relevant studies, based on a Web of Science Core Collection topic search for ''(''economic valu*'' OR ''monetary valu*'') AND soil*'' conducted in May 2019. Relevance for the review was assessed on the basis of the abstract. We considered only empirical papers reporting actual valuation studies (and not, e.g., conceptual papers on the topic) that had an explicit link to soils. We derived from the studies especially two types of information: which soil-based ecosystem services are valued (valuation objects), and which methods are used to estimate their economic value. A graphical analysis of the relationships between the ecosystem services addressed in the reviewed studies was conducted in R 3.5.1 (*R Core Team, 2018*), using the *igraph* package (*Csárdi, 2019*).

The results of the review are reported in ''Soil valuation studies: valuation objects and methods''. Before, we present a conceptual framework of soil-based ecosystem services.

## SOIL-BASED ECOSYSTEM SERVICES

Agricultural soils fulfil multiple important functions, namely the production of plant biomass, storing and filtering of water, storing and recycling of nutrients, habitat provision, and carbon storage (*Schulte et al., 2014*; *Vogel et al., 2018*) with the capacity to provide services to human societies (*De Groot et al., 2010*). The ''direct and indirect contributions of ecosystems to human well-being'' that are based on the use or appropriation of these soil functions are called ecosystem services (*Kumar, 2010*, p. xxxiv; see also *Boyd & Banzhaf, 2007*). Soil-based ecosystem services are the outcomes of soil processes that economic valuation focuses on in order to make visible the benefits of soils for human well-being and to inform sustainable soil management and policy. We build on the Common International Classification of Ecosystem Services (CICES V5.1) developed by the European Environment Agency (*Haines-Young & Potschin, 2018*). CICES provides a hierarchical structure that differentiates between abiotic and biotic ecosystem services and groups them into three sections: provisioning services, regulation & maintenance services, and cultural services. Each CICES section is further subdivided into divisions, groups and classes. In total, there are 84 classes and most assessments of ecosystem services are conducted at this level, though some studies also shift to higher hierarchical orders where assessment at class level is not feasible.

Soil-based ecosystem services are affected by soil management, and agricultural soils are subject to multiple forms of management. Agricultural soil management can be classified into four main categories: spatial cropping patterns, crops and rotations, mechanical pressures, and inputs into the soil (*Techen & Helming, 2017*). The typical soil management

practices within each category have different effects on ecosystem services (*Schwilch et al., 2018*). For instance, spatial cropping patterns influence below-ground as well as above-ground ecosystem services such as pest control through different degrees of habitat heterogeneity (*Rusch et al., 2016*). Mechanical pressures, such as the weight of machines and tillage, are mainly (but not exclusively) relevant below ground, for instance because they compact the soil and subsequently reduce crop yields and thus provisioning services (*Schjønning et al., 2015*).

Soils are multifunctional and it is not possible to simultaneously maximize the provision of all ecosystem services in one location. In agricultural soil management, trade-offs are unavoidable. They may occur between different services at the same place and time or across spatial and temporal scales, meaning that improvements in one service may imply deterioration of the same or another service at another location or with a certain time lag. This makes their analysis, assessment and valuation challenging. For example, at farm level, narrow crop rotations result in high provisioning services as food and feed production increase. However, at landscape level, the same management results in a low provision of habitats for farmland birds (*Gutzler et al., 2015*). Considering temporal scales, the application of fertilizer at intervals optimized for plant nutrition, irrespective of soil moisture conditions, results in high provisioning services in the short term. However, if farmers traverse their field with heavy machines during times when soils are too wet, this leads to in soil compaction, which reduces provisioning services in the long term (*Frelih-Larsen, Hinzmann & Ittner, 2018*).

The CICES framework is designed to assess services provided by all types of ecosystems rather than tailored to the specifics of agricultural soils. Therefore, some classes are very broad for the context of soils, e.g., class 2.2.4.2 *Decomposition and fixing processes and their effect on soil qualit* y, which encompasses the decomposition of biomass, nutrient cycling and nitrogen fixing by leguminous crops. Others are very specific, e.g., classes 3.1.2.3 and 3.1.2.4 that differentiate between aesthetically important and culturally important ecosystems. The partial overlapping of classes and the general complexity of the CICES framework preclude an intuitive understanding, which makes use of the framework in stakeholder communication difficult.

For most impact assessments, the number of 84 classes is too high to address. In the context of this paper, in which we address soil-based ecosystem services in agricultural production systems, we focus on the biotic ecosystem services. To derived a list of ecosystem services relevant for our analysis, we applied a two-step process. First, we eliminated services that are not provided by agricultural soils or not affected by agricultural soil management. This resulted in the removal of services provided by animals and such obtained from aquaculture or the gathering of material in the wild. Second, services that are uncommon in conventional agriculture in temperate zones or only provided within very specific settings were not considered. This led to the removal of multiple services, such as protection against avalanches and landslides, attenuation of smells and noises, or use for education. In this way, we reduced the number of biotic classes from 56 to 22 (Table 1). As the importance of spatial and temporal sensitivity both for soil processes (*Vogel et al., 2018*) and for soil assessment and governance has been emphasized (*Juerges, Hagemann & Bartke, 2018*), we

**Table 1 Common ecosystem services (based on CICES 5.1 classes) provided by agricultural soils in high-performance agricultural systems of the temperate zone, that are affected by agricultural management.** Short names used in text are bold. Columns "Main spatial scale" and "Main temporal scale" indicate the scale(s) at which each service is mainly provided.

| CICES Code | Provisioning Services | Main Spatial Scale | Main Temporal Scale |
|---|---|---|---|
| 1.1.1.1 | Cultivated terrestrial plants (including fungi, algae) grown for nutritional purposes [**Food production**] | Field | Short-term |
| 1.1.1.2 | Fibres and other **materials** from cultivated plants, fungi, algae and bacteria for direct use or processing (excluding genetic materials) | Field | Short-term |
| 1.1.1.3 | Cultivated plants (including fungi, algae) grown as a source of **energy** | Field | Short-term |

| CICES Code | Regulation & Maintenance Services | Main Spatial Scale | Main Temporal Scale |
|---|---|---|---|
| 2.1.1.1 | **Bio-remediation** by microorganisms, algae, plants, and animals | Field - Landscape | Short- to Mid-term |
| 2.1.1.2 | **Filtration/sequestration**/storage/accumulation by microorganisms, algae, plants, and animals | Field - Landscape | Short- to Mid-term |
| 2.2.1.1 | **Control of erosion** rates | Field - Landscape | Short- to Long-term |
| 2.2.1.3 | **Hydrological cycle** and water flow regulation (Including flood control, and coastal protection) | Landscape | Short- to Long-term |
| 2.2.2.1 | **Pollination** (or 'gamete' dispersal in a marine context) | Landscape | Short-term |
| 2.2.2.3 | **Maintaining nursery populations and habitats** (Including gene pool protection) | Landscape- international | Long- term |
| 2.2.3.1 | **Pest control** (including invasive species) | Field - Landscape | Short-term |
| 2.2.3.2 | **Disease control** | Field - Landscape | Short-term |
| 2.2.4.2 | **Decomposition and fixing** processes and their effect on soil quality | Field | Mid- to Long-term |
| 2.2.5.1 | Regulation of the chemical condition of freshwaters by living processes [**Regulation of freshwater chemistry**] | Landscape - International | Mid- to Long-term |
| 2.2.5.2 | Regulation of the chemical condition of salt waters by living processes [**Regulation of saltwater chemistry**] | Landscape - International | Mid- to Long-term |
| 2.2.6.1 | Regulation of chemical composition of atmosphere and oceans [**Climate regulation**] | Global | Long- term |
| 2.2.6.2 | Regulation of temperature and humidity, including ventilation and transpiration [**Microclimate regulation**] | Landscape | Mid- to Long-term |

| CICES Code | Cultural Services | Main Spatial Scale | Main Temporal Scale |
|---|---|---|---|
| 3.1.1.1 | Characteristics of living systems that that enable activities promoting health, recuperation or enjoyment through **active or immersive interactions** | Landscape | Short- to Mid-term |
| 3.1.1.2 | Characteristics of living systems that enable activities promoting health, recuperation or enjoyment through **passive or observational interactions** | Landscape | Short- to Mid-term |
| 3.1.2.3 | Characteristics of living systems that are resonant in terms of **culture or heritage** | Landscape | Short- to Mid-term |
| 3.1.2.4 | Characteristics of living systems that enable **aesthetic experiences** | Landscape | Short- to Mid-term |
| 3.2.2.1 | Characteristics or features of living systems that have an **existence value** | Landscape - Global | Short- to Long-term |
| 3.2.2.2 | Characteristics or features of living systems that have an **option or bequest value** | Landscape - Global | Short- to Long-term |

list for each service the spatial and temporal scales at which the service is mainly provided. Both the selection of services and the determining of main spatial and temporal scales are based on expert assessment by the authors.

# ECONOMIC VALUATION OF SOIL-BASED ECOSYSTEM SERVICES

One of the principal ways to assess the societal relevance of soil-based ecosystem services is by means of economic valuation. In this section, we first make some general remarks about the role of economic valuation and the type of information it provides ("Key elements of economic valuation in the soil context"). We then proceed to a review of soil valuation studies focusing on soil-based ecosystem services and the valuation methods used ("Soil valuation studies: valuation objects and methods"). On this basis, we want to highlight some gaps and the as-yet unrealized potential that economic valuation offers in the context of agricultural soils.

## Key elements of economic valuation in the soil context

The main rationale for using the economic valuation of environmental goods, including soil-based ecosystem services, is that it allows to compare welfare effects of changes in scarce goods that are not measured in a common metric. The resulting economic values can then be used for different policy-relevant purposes, such as communication of values, cost–benefit analysis and informing the setting of incentive payments (*Laurans et al., 2013*). For instance, a change in soil management (e.g., tillage) may lead to an increase in one ecosystem service (e.g., carbon storage) at the expense of another (e.g., food production). Economic valuation helps compare the two effects. The management and governance of multifunctional agricultural landscapes requires frequent decisions in trade-off situations. By facilitating comparison between different dimensions of a problem via expressing diverse impacts in one value unit (i.e., monetary terms), economic valuation is expected to help navigate such trade-offs. For instance, it may show that the production loss associated with applying minimum tillage is outweighed by the increase in soil water capacity (and thus water flow regulation and flood protection) (*Pereira et al., 2018*). Or it helps assess the trade-off between agricultural production on drained organic soils and the carbon storage in those soils when they are rewetted (see *Albert et al., 2017*).

Economic valuation has received a fair share of criticism (*Hausman, 2012*; *Baveye, Baveye & Gowdy, 2016*) and its results should be treated with caution and as one among many contributions to political decision making. Still, its role in many political processes is strong; so is the demand for monetary values to inform e.g., the design of agri-environmental payment schemes or the assessment of infrastructure projects with environmental consequences (*Förster et al., 2019*). Therefore, it is essential to identify valuation results of (relatively) high informational value and be aware of their limitations. This is particularly important in the context of soils and soil-based ecosystem services, where economic valuation is becoming increasingly widespread (see "Soil valuation studies: valuation objects and methods"). Here, we do not aim for a comprehensive introduction (for this, see *Pascual et al. (2010)*; in the context of soils, *Baveye, Baveye & Gowdy (2016)*). Rather, after briefly introducing the overarching concept of Total Economic Value (TEV) (*Krutilla, 1967*; *Pascual et al., 2010*), we highlight two fundamental aspects of economic valuation—the focus on incremental changes and the central role of preferences—that are

informative when making and analyzing choices with respect to valuation methods, an issue to which we will return in "Soil valuation studies: valuation objects and methods".

The most basic framework used in the economic valuation of environmental goods is Total Economic Value. Its main feature is the distinction between use values, associated with actually using the environmental good directly or indirectly, and non-use values, which do not require the environmental good being used by the valuer, i.e., the person ascribing value to the environmental good, herself. Furthermore, there are the additional categories of option value and insurance value, which have no agreed-upon "position" within the framework (*Pascual et al., 2010*). Especially the latter—insurance value—has become the focus of much conceptual research only recently, including in the context of soil valuation (*Pascual et al., 2015*). Figure 1 shows an extension of the original TEV with insurance value, based on *Pascual et al. (2015)* and *Bartkowski (2017)*. Here, the additional category of *uncertain-world values* encompasses values that arise when there is uncertainty over the future demand and supply of ecosystem services, namely option and insurance value, which are mainly attributable to (soil) biodiversity (*Bartkowski, 2017*; *Pascual et al., 2015*). Meanwhile, the conventional TEV categories can be found within *certain-world values*, i.e., they are untouched by and independent of considerations of uncertainty. Agricultural soils provide a range of direct and indirect use values; most soil-based provisioning services can be viewed as generating direct use values, while regulating services are usually associated with indirect use values. Moreover, any soil-based ecosystem service can also have altruistic or bequest value if it is viewed as benefiting others, while existence value is mainly attributable to objects, including soils as such.[1] Note that non-use values can be estimated only by means of stated preference methods; also, option value and insurance value are most easily captured by means of this method type (*Bartkowski, 2017*). This is the case because non-use values are by definition not expressed in market behaviour and cannot be traced back to market choices. Thus, one needs to apply hypothetical, survey-based valuation methods to obtain information on non-use values.

In addition to the TEV, there are two major characteristics of economic valuation that have particular relevance for the choice among and relative merit of the available valuation methods. First, economic valuation deals with incremental changes in the quality or quantity of a good's supply. It always implies a trade-off, a potential exchange of one scarce good against another. This, of course, presupposes substitutability—non-substitutable, essential goods cannot be sensibly expressed in terms of economic value (other than the value being infinite, see *Toman, 1998*; *Bockstael et al., 2000*). The centrality of the substitutability assumption also leads to the usual approach of estimating the economic value of incremental (best: marginal) changes in the quantity or quality of the good in question—the reasoning being that small changes in almost any good are substitutable, while larger changes may not be. Thus, economic valuation studies are usually based on the comparison of two marginally different states of the world (one usually being the status quo), for instance analyzing the preferences towards changes in soil-based ecosystem services resulting from a given change in agricultural management practices.

The second important and only seemingly trivial characteristic of economic valuation is that its goal is to provide a measure of preferences, i.e., subjective evaluations made

[1] In fact, CICES includes existence value and bequest value as individual ecosystem services, attributable to "characteristics or features of living systems" (see Table 1).

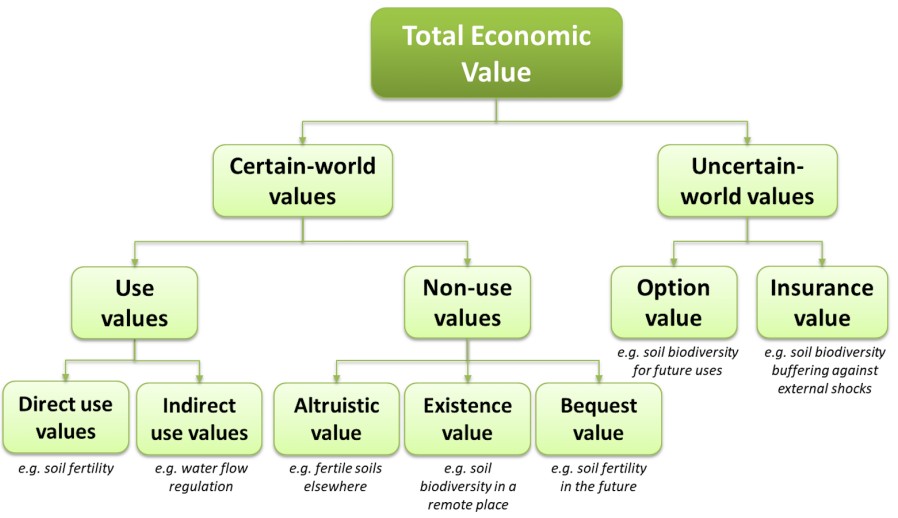

**Figure 1** **The Total Economic Value framework with insurance value.** Source: own elaboration based on *Bartkowski (2017)*.

[2]There is a long-standing debate whether observed preferences can be translated into utility, i.e., whether there is an intricate link between the two (*Samuelson, 1938*; *Sen, 1973*). This debate has repercussions for the theory of economic valuation (*Aldred, 1994*); here, we follow *Aldred (1994)* in restricting the domain of economic valuation to preferences, without linking it to the more problematic concept of utility.

by people, reflecting the relative scarcity of different goods and the trade-offs between them.[2] This implies, first, that economic valuation has an inherently anthropocentric focus—though non-anthropocentric considerations may and often do enter "through the back door" when they underlie the preferences observed among and expressed by valuing actors (*Spash, 2006*; *Martín-López, Montes & Benayas, 2007*). Second, and more importantly in the context of the present paper, a proper economic valuation study should be based on some way of observing or eliciting preferences. While other kinds of proxies are admissible in light of high resource demands and low precision requirements, economic valuation sensu stricto is an expression of preferences. These can be measured directly, by means of questionnaires eliciting preferences for hypothetical scenarios of change (stated preference methods), or indirectly by means of statistical analyses of observed behaviour in markets that can be linked to environmental goods (revealed preference methods). In the broad sense, the commonly used cost-based valuation methods, which generally approximate the economic value of an ecosystem or an ecosystem service by looking at the costs of its replacement, restoration, preservation etc., are seen as a class of economic valuation methods. However, they do not offer any insights into preferences—they can only tell us something about technical possibilities and the costs thereof. For instance, if the replacement cost approach is used to estimate the economic value of the flood protection service of a riparian landscape, one may use the cost of installing flood walls in downstream towns as a proxy. However, this may be both an underestimate (if people were actually willing to pay much more to preserve the flood protection service) and an overestimate (if they preferred the increased risk of flood to paying for the flood walls). Without additional information on preferences, we cannot know how accurate the proxy-based value is.

One could argue that the appropriateness of applying cost-based methods depends on whether the ecosystem service in question is in its essence a private good, such as

[3]This reasoning abstracts from the problem of myopic agents, who do not properly consider the consequences of their actions on themselves if these consequences are to be expected in the future—in other words, they discount these consequences heavily. This issue is particularly relevant in the case of agricultural soils, where tenancy has been shown to foster short-term oriented management by the tenant (*Soule, Tegene & Wiebe, 2000*; but see *Leonhardt, Penker & Salhofer, 2019*). Here, however, it may be argued that the case is similar to public goods—the trade-off implicit in the use of a cost-based proxy does not reflect the calculus of a single agent, but rather of two agents (tenant and owner).

e.g., food production, or a public good, such as e.g., nutrient cycling (see *Bartkowski et al., 2018*). The reason is that the costs that are used as proxy would indeed be borne by the person managing the land/soil and thus "responsible" for changes in the ecosystem service. Therefore, using a cost-based proxy essentially reflects the soil manager's calculus. Even though it still does not reflect her preferences, the two implicitly compared options (management for or the loss of the ecosystem service) are indeed the options faced by the same person.[3] On the other hand, for public ecosystem services, the use of cost-based proxies is more problematic, as here the implicit trade-off does not reflect the calculus of an individual agent, but rather of two different (groups of) agents. In such a situation, information on actual preferences is much more essential for guiding decisions. Since most ecosystem services, including soil-based ones, are public goods, this still implies that the application of cost-based valuation methods is problematic.

Of course, applying preference-based valuation methods does not per se lead to valid results. It is advisable to follow guidelines for the application of such methods, as outlined e.g., in *Johnston et al. (2017)* for stated preference methods and in *Riera et al. (2012)* for both revealed and stated preference methods. Even then, given the complexity of ecosystems and the many factors influencing human decision making, results of economic valuation studies should be interpreted primarily as giving orientation about orders of magnitude and value ranges (see *Förster et al., 2019*), rather than offering precise values (we will come back to this in "Economic valuation for sustainable soil management and policy"). Still, following guidelines such as those referred to above and making use of advances in study design and analysis, precision and relevance for political purposes can be increased.

## Soil valuation studies: valuation objects and methods
### Overview
Table 2 provides a list of the studies included in the review, together with the year of study (where applicable and identifiable), study area and a categorization into studies estimating total versus marginal values of soil-related ecosystem services.

Among the 43 studies included here, 13 were conducted in North America (US or Canada), 12 in Europe and nine in Asia.

[4]It should be noted that this is to some extent true for many soil-based ecosystem services, e.g., carbon storage, which has an influence on other ecosystem services as well. In fact, interdependencies between ecosystem services are the norm rather than an exception (*Cord et al., 2017*). However, soil erosion is "special" as it matters only because of the other ecosystem services it affects.

### Ecosystem services studied
The coverage of soil-based ecosystem services in soil valuation studies is quite uneven. Most of them focus on a handful of ecosystem services (particularly Climate regulation and Decomposition and fixing) whose value can be estimated relatively easily. Also, there exist multiple studies looking at the (social) cost of soil erosion. They can be interpreted as valuations of the CICES ecosystem service Erosion control. But soil erosion may also be interpreted as negatively affecting a bundle of soil-based ecosystem services—most of the papers in this context do not, however, disentangle the different ecosystem services and benefits affected by the loss of topsoil.[4] Conversely, there are hardly any studies investigating cultural ecosystem services provided by soils. In what follows, we briefly discuss the most widely valued soil-based ecosystem services. For an overview of the estimate ranges across different studies, see *Jónsson & Davíðsdóttir (2016)*. The combinations of particular

**Table 2   Overview of publications included in the review.**

| Code | Study publication | Year of valuation | Study location | Type of valuation |
|---|---|---|---|---|
| 1 | *Miranowski & Hammes (1984)* | 1978 | Iowa, US | Marginal |
| 2 | *Moore & McCarl (1987)* | NA | Willamette Valley, Oregon, US | Marginal |
| 3 | *Pimentel et al. (1995)* | NA | global | Total |
| 4 | *Pimentel et al. (1997)* | NA | US | Total |
| 5 | *Pretty et al. (2000)* | 1996 | UK | Total |
| 6 | *Colombo, Calatrava-Requena & Hanley (2003)* | NA | Alto Genil basin, Spain | Marginal |
| 7 | *Tegtmeier & Duffy (2004)* | 2002 | US | Total |
| 8 | *Xiao et al. (2005)* | NA | Shanghai, China | Total |
| 9 | *Colombo, Calatrava-Requena & Hanley (2006)* | NA | Alto Genil basin, Spain | Marginal |
| 10 | *Decaëns et al. (2006)* | NA | NA | Total |
| 11 | *Hansen & Hellerstein (2007)* | 1997 | US | Marginal |
| 12 | *Sandhu et al. (2008)* | NA | Canterbury, New Zealand | Total |
| 13 | *Porter et al. (2009)* | NA | Taastrup, Denmark | Total |
| 14 | *ChoCho & Rapera (2010)* | 2004 | Inle Lake Watershed, Myanmar | Total |
| 15 | *Bond, Hoag & Kipperberg (2011)* | NA | Northeastern Colorado, US | Marginal |
| 16 | *Glenk & Colombo (2011)* | 2008 | Scotland, UK | Marginal |
| 17 | *Kiran & Malhi (2011)* | 2009 | Halol Range, Gujarat, India | Non-marginal change |
| 18 | *Mekuria et al. (2011)* | NA | Tigray, Ethiopia | Total |
| 19 | *Almansa, Calatrava & Martínez-Paz (2012)* | NA | Aljibe Basin, Spain | Total |
| 20 | *Rodríguez-Entrena et al. (2012)* | 2011 | Andalusia, Spain | Marginal |
| 21 | *Bastian et al. (2013)* | NA | Görlitz, Germany | Total |
| 22 | *Lu et al. (2013)* | NA | Qinghai–Tibet Plateau, China | Total |
| 23 | *Samarasinghe & Greenhalgh (2013)* | 2007 | Manawatu catchment, New Zealand | Marginal |
| 24 | *Alam et al. (2014)* | NA | NA | Total |
| 25 | *Dominati et al. (2014b)* | NA | Hawke's Bay, New Zealand | Total |
| 26 | *Dominati et al. (2014a)* | NA | Waikato region, New Zealand | Total |
| 27 | *Dechen et al. (2015)* | 1996 | Campinas, Brazil | Total |
| 28 | *Fan, Henriksen & Porter (2016)* | 2015 | Taastrup, Denmark | Total |
| 29 | *Jerath et al. (2016)* | NA | Everglades, Florida, US | Total |
| 30 | *Noe et al. (2016)* | 2010 | Minnesota, US | Total |
| 31 | *Hungate et al. (2017)* | NA | Cedar Creek, Minnesota, US | Marginal |
| 32 | *Kibria et al. (2017)* | NA | Veun Sai-Siem Pang National Park, Cambodia | Total |
| 33 | *Levykin et al. (2017)* | NA | Orenburg, Russia | Total |
| 34 | *Liu et al. (2017)* | NA | Sanjiang Plain, China | Total |
| 35 | *Bashagaluke et al. (2018)* | 2017 | Anwomaso, Kumasi, Ghana | Total |
| 36 | *Campbell (2018)* | NA | Maryland, US | Total |
| 37 | *Cerda et al. (2018)* | 2013 | Llanos de Challe, Chile | Marginal |
| 38 | *Ganguly et al. (2018)* | NA | Palk Bay and Chilika, India | Total |
| 39 | *Hopkins et al. (2018)* | NA | Difficult Run watershed, Virginia, US | Total |
| 40 | *Lee et al. (2018)* | 2010 | Korea | Total |
| 41 | *Mastrorilli et al. (2018)* | NA | Bonis basin, Italy | Total |
| 42 | *Kay et al. (2019)* | NA | Multiple in Europe | Total |
| 43 | *Plaas et al. (2019)* | 2017 | Lower Saxony, Germany | Total |
**Table 3  Application of valuation methods for particular soil-based ecosystem services in analyzed studies (study codes see Table 2).**

| Ecosystem service (CICES class)/Method | Market price | Cost-based | Revealed preferences | Stated preferences |
|---|---|---|---|---|
| Food production, materials or energy | 10, 12, 25, 26, 33 | | | |
| Bio-remediation or filtration/sequestration | | 25, 26 | | |
| Control of erosion | | 5, 14, 19, 21, 22, 32, 36, 40, 41 | 1 | 6, 15, 19, 37 |
| Hydrological cycle | 42 | 2, 3, 7, 11, 25, 26, 28, 32, 34, 41 | 23 | |
| Pest control | | 25, 26, 43 | | |
| Regulation of freshwater chemistry | | 2, 7, 12, 39 | | 9 |
| Climate regulation | 8, 12, 13, 25, 26, 42 | 5, 8, 22, 28, 29, 30, 31, 32, 38 | | 16, 20 |
| Decomposition and fixing | 13, 24 | 3, 4, 12, 13, 14, 17, 18, 22, 25, 26, 27, 28, 32, 35 | | |
| Active interactions | | 7 | | |
| non-CICES Physical environment | 25, 26 | 25, 26 | | |

valuation methods and specific soil-based ecosystem services found in the literature are depicted in Table 3 and will be critically discussed further below.

The most comprehensive economic valuation studies of soil-based ecosystem services were conducted by Dominati and colleagues in New Zealand (*Dominati et al., 2014a*; *Dominati et al., 2014b*). Using predominantly cost-based valuation methods, they analyzed a number of soil-based ecosystem services in different landscapes, including: (agricultural) biomass production (CICES: Food production, Materials, Energy), physical support of animals and infrastructure, flood mitigation (Hydrological cycle), nutrient cycling (Decomposition and fixing), climate regulation, and pest control. A network graph depicting the co-occurrences of the different ecosystem services across studies can be found in Fig. 2. *Dominati et al. (2014b)* found that nutrient cycling and flood mitigation are responsible for the largest share of soils' total economic value. In a pastoral agricultural landscape in New Zealand, *Dominati et al. (2014a)* found that the value of soil-based regulating services is about 2.5 times as high as that of soil-based provisioning services. In this case, the services with the highest value were what the authors called filtering of nutrients and contaminants (Bio-remediation/Filtration/Sequestration), followed by the provision of food (Food production) and flood mitigation (Hydrological cycle). A similarly comprehensive approach can be found in *Porter et al. (2009)* and *Sandhu et al. (2008)*, who included a number of soil-based ecosystem services (nitrogen regulation, soil formation, soil carbon/carbon regulation, hydrological flow) in their comparisons of different types of agriculture in Denmark and New Zealand, respectively.

Other studies had a much narrower focus, usually on single soil-based ecosystem services. We ignore here those soil-based provisioning ecosystem services, such as food, fibre and raw materials, whose economic value can be derived or at least approximated by means of market prices (see *Jónsson & Davíðsdóttir, 2016*; *Robinson et al., 2014*).

*Water-related ecosystem services.* The contribution of soils to water quality and fresh water provision (Regulation of freshwater chemistry) can be approximated by means of the replacement cost method—how much does it cost to clean water for drinking (i.e., if

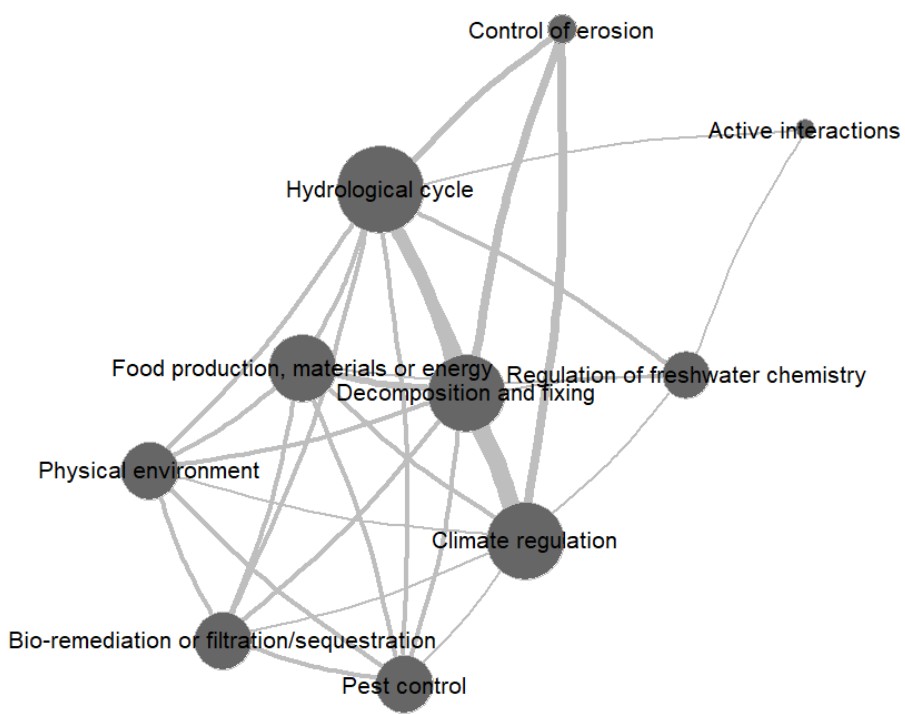

**Figure 2** **Network graph of ecosystem service co-occurrences across reviewed valuation studies (source: own elaboration).** Vertix sizes indicate number of occurrences of an ecosystem service; edge widths correspond to frequency of co-occurrence.

cleaning is not "done" by soils)? This approach has been applied by *Tegtmeier & Duffy (2004)*. Another possible approach, chosen for instance by *Dominati et al. (2014a)* and *Dominati et al. (2014b)*, is to look at nutrient retention by soils more directly, and value it by means of methods such as provision costs and avoidance costs (in the case of nutrient leaching). A soil-based ecosystem service that is frequently studied in valuation studies is the hydrological cycle, including flood protection (mostly cost-based methods), see e.g., *Hansen & Hellerstein (2007)* or *Tegtmeier & Duffy (2004)*. They use the costs of alternative measures of flood protection to approximate the economic value of soils' capacity to store water. In a more direct approach, *Kay et al. (2019)* used water prices for this soil-based ecosystem service, interpreting it as groundwater recharge.

*Climate regulation.* The economic value of carbon sequestration in soils (Climate regulation) can be estimated seemingly easily on the basis of the various estimates of social cost of carbon (SCC) available in the climate economics literature (*Van den Bergh & Botzen, 2015*) or using carbon prices in emissions trading schemes. The latter approach, however, is problematic, as these prices usually do not reflect the marginal damage costs of carbon in terms of its contribution to climate change, but rather are a result of political decisions such as the cap level, the method of the first distribution of emission certificates (e.g., grandfathering or auctioning), leakage effects due to the coverage of only some sectors of the economy etc. (*Hintermann, Peterson & Rickels, 2016*). SCC estimates are

usually generated by means of integrated assessment models (IAM). The estimates are very sensitive to a number of model parameters (*Ackerman et al., 2009*), especially the so-called damage function, which links changes in global temperature to losses in terms of capital, production, human lives etc., and the social rate of discount, which allows the comparison of effects occurring at different points in time, and which enticed a large and controversial literature of its own (*Arrow et al., 2014*). As a result, SCC estimates vary in the range of orders of magnitude (*Van den Bergh & Botzen, 2015*) it is by no means clear which estimate to use when valuing, e.g., carbon storage by soils. A noteworthy study using this approach is *Hungate et al. (2017)*, who used SCC estimates to value the contribution of soil biodiversity to carbon storage. An alternative to the use of SCC estimates is the approach by *Rodríguez-Entrena et al. (2012)*, who conducted a choice experiment to evaluate the demand for carbon sequestration in olive grove soils in Andalusia (Spain) and came up with a willingness to pay by the general public of 17 EUR per ton $CO_2$ per person—as compared to a range of 5 to 106 USD in studies using the most common IAMs (*Van den Bergh & Botzen, 2015*). A similar approach, though with a focus on ancillary effects of soil carbon management, was used by *Glenk & Colombo (2011)* in Scotland. Another study worth noting in this context is *Noe et al. (2016)*, who used a Monte Carlo analysis of SCC estimates to identify the value of carbon storage in Minnesota prairies; they found an average value of 73 USD per ha per year. In another study, *Jerath et al. (2016)* estimated the economic value of carbon storage in the Everglades (US), showing that carbon storage in soils amounts to between 77 and 90 per cent of the overall value across study sites. *Lu et al. (2013)* used a rather unconventional approach for valuing climate regulation by soil carbon storage, namely approximating it by the costs of planting trees that would bind an equivalent amount of carbon.

*Cultural ecosystem services.* The economic valuation of cultural ecosystem services is a challenge even beyond the context of soils (*Baveye, Baveye & Gowdy, 2016*; *Chan et al., 2012*). Valuation studies of soil-based cultural ecosystem services are scarce; the only one cited by *Jónsson & Davíðsdóttir (2016)* in their comprehensive literature review is the conference paper by *Eastwood, Krausse & Alexander (2000)*, who estimated the economic cost of soil erosion in terms of recreational loss in New Zealand. A similar approach can be found in *Tegtmeier & Duffy (2004)*, who looked at the damage costs of sedimentation in rivers (caused by agricultural soil loss) in terms of foregone river-based recreation. In their study of the economic value of erosion control, *Almansa, Calatrava & Martínez-Paz (2012)* mention a few "benefits" of erosion control that can be interpreted as cultural ecosystem services (e.g., "increase in aesthetic and recreational use", "rural tourism"). However, as they elicit willingness to pay (WTP) for an erosion control programme as a whole, they cannot distinguish between the relative contributions of the various benefits to the overall WTP.

*Other ecosystem services.* We did not find valuation studies looking at the soil-based ecosystem services Maintaining habitats, Disease control and microclimatic regulation (Regulation of temperature/humidity/ventilation/transpiration) or most cultural ecosystem

services—despite their importance for the overall picture of soils' contributions to human well-being (*Helming et al., 2018*; *Van der Meulen et al., 2018*). Of those soil-based ecosystem services that have actually been recognized in valuation studies, many have been included only infrequently. For instance, Chemical regulation of freshwaters—a crucial ecosystem service related to the nexus connecting agriculture with aquatic ecosystems—can only be found in five studies. Pest control, highly important given the agronomic relevance of soil-living pests (e.g., *Kulmatiski et al., 2014*), occurs in only three soil valuation studies. Soil-based cultural ecosystem services are almost completely missing, while the economic value of the non-CICES ecosystem service/soil function Physical environment (for human structures, housing, livestock etc.) has only been studied by *Dominati et al. (2014a)* and *Dominati et al. (2014b)*.

### Valuation methods

Table 3 above summarizes the application of valuation methods in studies estimating the economic values of different soil-based ecosystem services. Most soil valuation studies use cost-based methods or market price proxies to estimate the economic value of soil ecosystem services. Both approaches have limits—as mentioned in "Soil-based ecosystem services", cost-based methods are easy to use but inconsistent with economic theory. Furthermore, the impact of the technical solutions whose costs are used as proxy of ecosystem service value on other ecosystem services are usually not taken into account. Market price proxies (e.g., the price of topsoil, *Robinson et al. (2014)*) are also problematic. In most cases, the market good (e.g., topsoil) is not equivalent to any soil-based ecosystem service, so its price is only a very rough proxy. Moreover, market prices are usually distorted due to imperfect markets (caused by taxes/subsidies, market power etc.). Thus, such approaches can be helpful as a first estimate, but their informational quality is limited in most cases. However, as Table 3 shows, they are quite common in the context of valuing soil-based ecosystem services.

Theoretically, land prices should reflect, among other factors such as proximity to public infrastructure, the value of soil-based ecosystem services, at least those that directly benefit landowners. Thus, "one would think that it would be feasible to disaggregate land prices into the prices of the various below- and above ground components of land, and eventually to estimate the monetary value of soils" (*Baveye, Baveye & Gowdy, 2016*, p. 28). As it turns out, however, the actual disentanglement of the relative contributions of the relevant factors to the price of land, is anything but straightforward and simple. Hedonic pricing is the standard approach here, which consists in a statistical analysis of various factors influencing land or real estate prices. In a rare instance of such an analysis of land prices, *Samarasinghe & Greenhalgh (2013)* used a hedonic pricing approach to determine the influence of inherent characteristics of soils on farmland prices in New Zealand. However, they did not explicitly value soil-based ecosystem services. Furthermore, such an approach would only allow for the estimation of private-good type soil-based ecosystem services. Public benefits are not likely to be reflected in land prices. Even private benefits such as the yield potential of soils are not necessarily related to land prices (*Daedlow, Lemke & Helming, 2018*).

There are only a few instances of stated preference methods being applied in the soil valuation context. For example, the already mentioned study by *Almansa, Calatrava & Martínez-Paz (2012)* analyzes the social benefits of erosion control by comparing restoration costs and contingent valuation—they find that the latter results in WTP estimates around twice as high as the former. *Rodríguez-Entrena et al. (2012)* also focus on erosion and erosion control in olive groves while using a choice experiment. However, they find negligible WTP for erosion control. *Colombo, Calatrava-Requena & Hanley (2006)* use a choice experiment to estimate the WTP for a reduction of off-site consequences of soil erosion. A choice experiment was also used in the already mentioned study by *Glenk & Colombo (2011)*. *Cerda et al. (2018)* include in their choice experiment an attribute *soil quality*, which cannot be easily linked to any specific soil-based ecosystem service.

Overall, Table 3 highlights two common features: (i) a strong bias in favour of the easiest-to-use, fairly transparent and easily reproducible, but also most imprecise and theoretically problematic cost-based methods; and (ii) a rather strong focus on a small number of particularly easy-to-quantify soil-based ecosystem services that can be easily valued, especially Decomposition and fixing (usually valued by means of the proxy of fertilizer costs) and Climate regulation via carbon storage (valued using SCC estimates or market prices). An exception are the many soil erosion control studies, as soil erosion affects many different soil-based ecosystem services across spatial and temporal scales. However, except for *Pimentel et al. (1995)*, none of them differentiates between the various elements of the ecosystem service bundle affected by soil erosion. Even in the few studies that considered more than one soil-based ecosystem service (e.g., *Pimentel et al., 1995*; *Dominati et al., 2014a*; *Dominati et al., 2014b*; *Sandhu et al., 2008*; *Porter et al., 2009*), only cost-based valuation methods were used and for each ecosystem service, the economic value was estimated separately. Meanwhile, more advanced valuation methods such as hedonic pricing or choice experiments allow for estimating the WTP for multiple ecosystem services simultaneously, reflecting trade-offs between them. Furthermore, we do not find any apparent changes over time with respect to both ecosystem service selection and choice of valuation methods.

## ECONOMIC VALUATION FOR SUSTAINABLE SOIL MANAGEMENT AND POLICY

The economic valuation of soil-based ecosystem services has the potential to ensure public awareness on societal importance of soils, including agricultural soils. It can do so by emphasizing that the contributions of soils to human well-being—via food production, flood protection and other effects—are valuable despite not having a market price. Furthermore, it can show how valuable contributions of soils are affected by soil management—and thus inform agricultural, environmental and other policies relevant for soils (e.g., *Glæsner, Helming & De Vries, 2014*; *Turpin et al., 2017*; *Vrebos et al., 2017*). Moreover, the economic valuation of soil-based ecosystem services may offer pathways for developing policy instruments, such as agri-environmental payments for management practices that promote soil-based ecosystem services. However, as shown in the review in
the previous section, the economic valuation of soil-based ecosystem services has not yet lived up to this potential for multiple reasons. In the following, we will discuss where we see untapped potentials, but also point to limitations of economic valuation in the context of soils.

Most obviously, the widespread use of cost-based valuation methods should be viewed critically for reasons mentioned in "Soil-based ecosystem services". They can be helpful in providing first rough estimates—but economics has to offer much more sophisticated valuation methods that are more informationally rich. While stated preference methods have been criticized because of validity and reliability problems (e.g., *Hausman, 2012*; *Rakotonarivo, Schaafsma & Hockley, 2016*), proper design following standard guidelines helps avoid many of the problems involved (*Riera et al., 2012*; *Johnston et al., 2017*; *Bishop & Boyle, 2019*).

It is important to emphasize that for economic values to be meaningful, their estimation should be based on environmental/land-use changes—while many existing soil valuation studies rather focus on the economic value of states, without explicit reference to (scenarios of) change (see Table 2). Especially in policy contexts, however, information about the economic value of changes in the provision of soil-based ecosystem services is crucial (see *Förster et al., 2019*). For instance, if economic values are to inform the setting of payment levels for result-based agri-environmental schemes (*Burton & Schwarz, 2013*), it is necessary that they can be expressed in terms of the change that the scheme is meant to incentivize. In any case, informing the setting of incentives requires high-quality valuation studies conducted specifically for this purpose; a direct transfer of economic values from one study to another context would be problematic.

There are also clear challenges involved in the economic valuation of soil-based ecosystem services generally, beyond the use of particular methods and focus on particular ecosystem services. For instance, the relevant temporal and spatial scales are a challenge. As indicated in Table 1, different soil-based ecosystem services are relevant at different temporal scales —thus, the same environmental and management change will be "felt" in terms of ecosystem service provision at different points in time. This implies that different relevant stakeholders, who weigh soil-based ecosystem services differently, will have different preferences with respect to a given soil management or land-use change. Here, economic valuation has both potential and an important limitation. On the one hand, it requires the identification of all relevant stakeholders (*Pascual et al., 2010*), which is helpful here. On the other hand, economic valuation is based on the Kaldor-Hicks criterion, i.e., it does not distinguish between winners and losers, but focuses on aggregate preferences. Therefore, to distinguish heterogeneous preferences resulting from the different time scales involved in soil-based ecosystem services, additional, complementary methods are necessary. Another possibility would be to apply economic valuation, but to distinguish between the WTP of different groups (*Cavender-Bares et al., 2015*).

With respect to the spatial dimension, an economic valuation study should be based on a proper understanding of the particular systemic, emergent effects of land-use changes from which economic values are to be derived. Here, multiple challenges arise. First, the ex-ante assessment (e.g., by means of modelling, see *Vogel et al., 2018*) of the specific effects

of a given land-use change on soil functions and soil-based ecosystem services involves many uncertainties and is very case-specific. Recently, novel design approaches to deal with ecosystem service uncertainty in stated preference studies have been developed (*Czajkowski, Hanley & LaRiviere, 2016*; *Faccioli, Kuhfuss & Czajkowski, 2019*), which explicitly include uncertainty in the scenarios to be evaluated by respondents (e.g., as attributes in a choice experiment). Second, there is a need to scale up the results in order to translate them to societally relevant spatial levels, usually landscapes. Here, combining effects of land-use or soil-management changes on soil functions with other, non-soil factors adds to the complexity. For instance, it is non-trivial to find adequate indicators linking spatial levels. However, it is crucial to understand the trade-offs involved at both (stylized) levels. The spatial dimension is also crucial in a more general sense, as "multiple authors have demonstrated or argued that the relevance of spatial patterns for policy evaluation can outweigh comparable effects of statistical or methodological issues that are often given greater attention in the literature [on economic valuation]" (*Glenk et al., 2020*, p. 216). For example, the geographic location of different groups of stakeholders/beneficiaries vis-à-vis the spatial extent of different soil-based ecosystem services is relevant and adds to the complexity mentioned above with respect to different time scales (*Budziński et al., 2018*). Also, soil management is spatially specific, so it may trigger different changes in the provision of soil-based ecosystem services across the landscape; their consequences for human well-being and the associated economic values depend on spatial interdependencies and patterns as well as neighbourhood effects. So far, even though the spatial nature of many soil-related ecosystem services is rather obvious, little attention has been given to its relevance for economic valuation. Approaches and methods that help to accommodate these considerations are available. They need to be taken up in the context of soil-based ecosystem service valuation.

Estimating the *relative* contribution of soils to the provision of most ecosystem services (and, thus, to their economic value) is a problem that is particularly difficult to resolve. For some ecosystem services, isolation of the contribution of soils may be possible, for example climate regulation via carbon storage. Yet, for many others, the interactions behind the provision of ecosystem services are way too complex—in terms of system understanding, but also when it comes to definitions distinguishing between soil and other (related) factors. Rather, it is the case that soil functions contribute to multiple ecosystem services, but they are usually not identical—other factors play an important role as well. For instance, what is the soil's contribution to the economic value of food production as compared to above-ground vegetation? What about soil's capacity to store water as contribution to flood protection? The question is, however, what the point of such "distilling" the relative contribution of soils may be. For policy purposes, it seems rather irrelevant—here, the relevant *leverage points* are above all management practices, which are not limited to affecting soils only. A related issue is the widely discussed (*Boyd & Banzhaf, 2007*) close interaction between natural and human contributions to the provision of ecosystem services. In the context of agriculture, the provision of soil-based ecosystem services strongly depends on the interaction between biogeochemical processes and human management. For example, the provision of food from an agricultural field relies on

human management such as the sowing of seeds and on natural factors such as rainfall. Since it is the (soil-based) ecosystem services which are valuable for human well-being, it is relevant how they (and their economic value) respond to changes in management, while the relative contribution of soils to these responses is, if anything, of purely academic interest. Furthermore, for policy purposes, it is essential to identify and take into account changes in the economic value of all affected ecosystem services when the consequences of a change in management, land use or other boundary conditions are investigated.

Overall, to be informative and useful for various policy purposes, the economic valuation of soil-based ecosystem services should make use of state-of-the-art approaches and methods. Particularly, we emphasize the importance of preference-based methods that allow for disentangling various attributes (ecosystem services) and their relative values (choice experiments, hedonic pricing). Moreover, these methods should be applied in line with established guidelines (*Riera et al., 2012*; *Johnston et al., 2017*; *Bishop & Boyle, 2019*). Another important point of particular relevance to the valuation of soils is taking into account spatial heterogeneity by means of appropriate methods and approaches (for an overview, see *Glenk et al., 2020*). Last but not least, here and elsewhere, the limitations of economic valuation should be acknowledged (see *Baveye, Baveye & Gowdy, 2016*; *Pascual et al., 2010*) and the application of hybrid or non-monetary methods should be considered where these limitations are particularly severe, e.g., in the case of cultural soil-based ecosystem services (see *Christie et al., 2012*; *Hattam et al., 2015*; *Lienhoop, Bartkowski & Hansjürgens, 2015*).

## RELEVANCE OF THE ECONOMIC VALUATION OF SOIL-BASED ECOSYSTEM SERVICES FOR POLICY MAKING

In this paper, we provided a comprehensive review of existing soil valuation studies with a focus on the ecosystem services addressed as well as the valuation methods applied. We found that preference information on soil-based ecosystem services (as provided by valuation studies) is scarce and that a small number of services is overrepresented (particularly those related to soil carbon storage and nutrient cycling). While there exist valuation approaches that would help improve the quality of information provided by soil valuation studies, they have not yet been applied in this context. Most existing studies rely heavily on rather imprecise and theoretically problematic cost-based methods.

For economic valuation to inform decisions and guide them, the quality of soil valuation studies needs to improve. Currently, the required level of quality is not given and the informational value of existing valuation results is low. In particular for services relating to aesthetic, spiritual or cultural values, reliable valuation methods are missing. Future research will have to elucidate in how far economic valuation is suitable to represent those services and whether other, hybrid or non-monetary methods are more appropriate for this purpose. More generally, economic valuation needs to be complemented with other perspectives better reflecting values related to cultural ecosystem services, while at the same time accounting for diverse issues such as equality, legitimacy or human health (see e.g., *Brevik et al., 2019*).

For many other services, however, the application of established, state-of-the-art methods could make economic valuation a valuable tool to support political decision making—especially so if existing approaches that allow to account for the heterogeneity and multifunctionality of soils are drawn upon. In what follows, we want to shed some light on the potential role of economic valuation in policy-making processes while applying the concept of a policy cycle.

As stated in the introduction, economic values can be policy relevant in different ways, being informative, decisive, and/or technical (*Laurans et al., 2013*). Thus, the economic valuation of soil-based ecosystem services has implications at various levels and purposes for land-use and soil-related policies. Following *Wegrich & Jann (2007)*, four phases of a typical policy-making process can be distinguished: (1) agenda setting, (2) policy formulation and decision making, (3) legitimation and implementation, and (4) evaluation and termination. At each stage, economic valuation can potentially contribute, though given the currently poor state of the empirical literature reviewed above, its potential is still largely untapped. In the following, we point to the requirements future soil valuation studies should fulfil to become relevant for the different stages of the policy cycle (based on the discussion in the previous section).

In the context of the agenda setting phase of the policy cycle, economic valuation has been used to emphasize market failures and the need for state intervention (*Bromley, 1996*; *Hubacek & Van den Bergh, 2006*; *Bartkowski et al., 2018*) by pointing out the non-market value of soil-based ecosystem services. As *Braat (2013)* emphasizes, valuation processes can be interpreted "as a form of regulatory adaptation via positive and negative feedback in an economic system. In this view, the valuation of changes in biodiversity, natural capital, and ecosystem services becomes a logical and necessary element of a sustainable development policy cycle" (p. 101). In the context of agenda setting, precision and state-of-the-art methodologies do not play a large role; however, a comprehensive set of values for different ecosystem services is necessary for a broad picture of soils' societal importance, which may need to go beyond economic values.

When it comes to policy formulation, economic valuation can contribute important information needed to set the policy objective at an effective aim—i.e., to clarify which end shall be achieved by an intervention. On the strategic level, valuation of soil-based ecosystem services can help decision makers on the local to global level design instruments fostering sustainable soil management. The multifunctionality of soils is the major challenge for target setting (effectiveness) before determining the means to get there (efficiency). In the context of policy formulation, precise estimates based on state-of-the-art methods are crucial. Also, they should be available for the full suite of soil-based ecosystem services. Uncertainties regarding the underlying biophysical phenomena and the valuation itself should be clearly communicated (e.g., it is not sensible here to work with single values, but rather with value ranges or indications of the level of confidence on which a given value estimation is based, following standards established in the context of the IPCC, see *Helgeson, Bradley & Hill, 2018*).

With respect to the legitimation phase of the policy cycle, it should be emphasized once more that low-quality estimates of the economic value of soil-based ecosystem services

can have detrimental effects in terms of legitimacy of the policies they inform (see *Droste & Meya, 2017*). This strengthens the call of this paper for better, state-of-the-art and comprehensive valuation studies in the soil context.

Economic valuation can be a useful instrument to assess if policies are efficient and achieve the desired effects. However, it has also been pointed out that policy makers often follow short-term, cost—benefit paradigms to avoid short-term costs for voters and stakeholders (e.g., *Göpel, 2016*). Transparent assessments can help compare and consider short- and long-term impacts. Yet, the application of economic valuation in policy making is still not common—in particular, more complex modelling and monetization methods are rarely applied (*Turnpenny et al., 2015*) —and is challenged by temporal and spatial sensitivity and dispersion of decisions and impacts. A typical problem faced by governance of natural resources is the spatial mismatch of ecological functional units and geographical policy-relevant units related to jurisdictions or property owners (*Cash et al., 2006*; *Leventon et al., 2019*). As pointed out by *Juerges, Hagemann & Bartke (2018)*, soil policies in administrative boundaries have to consider ecological boundaries, too. Given the complexity of the soil system (*Vogel et al., 2018*) and the soil challenges (*Nathanail et al., 2018*), an assessment at multiple spatial scales is required (*Artmann, 2015*). To be fruitful, economic valuation is needed at the (spatial) scale referring to the policy objective (agenda setting) with the respective time scales and inherent value system (*Campbell et al., 2001*). In this context, it is essential to use economic valuation in combination with other sources of information in a holistic, systemic manner (*Vogel et al., 2018*): for instance, ex-ante evaluation of changes in soil policy and management can be assessed by the combination of systemic process modelling with economic values to find out whether the changes have had welfare-increasing effects.

Overall, our review and the following analysis of the potential role of the economic valuation of soil-based ecosystem services in land-use policy suggest that while economic valuation can be useful and informative, the current status of the literature in this area is unsatisfactory and needs significant improvement. We showed that state-of-the-art preference-based valuation methods and recent developments that take into account spatial effects and uncertainty could improve the quality of value estimates for soil-based ecosystem services. Further, we highlighted which role economic valuation can play at various stages of soil policy making. First, however, there is an urgent need for improvement in methodological rigour in this area.

## ACKNOWLEDGEMENTS

We would like to thank Victoria Dietze for assistance with literature search, Anne Wessner for a language check, as well as Julian Massenberg and the two reviewers for valuable comments. The usual disclaimer applies.

### Funding

This work was funded by the German Federal Ministry of Education and Research (BMBF) in the framework of the funding measure "Soil as a Sustainable Resource for the Bioeconomy—BonaRes", project "BonaRes (Module B): BonaRes Centre for Soil Research, subproject A" (grant 031B0511A). The funders had no role in study design, data collection and analysis, decision to publish, or preparation of the manuscript.

### Grant Disclosures

The following grant information was disclosed by the authors:
German Federal Ministry of Education and Research (BMBF): 031B0511A.

### Competing Interests

The authors declare there are no competing interests.

### Author Contributions

- Bartosz Bartkowski conceived and designed the experiments, performed the experiments, analyzed the data, prepared figures and/or tables, authored or reviewed drafts of the paper, and approved the final draft.
- Stephan Bartke, Katharina Helming, Anja-Kristina Techen and Bernd Hansjürgens analyzed the data, authored or reviewed drafts of the paper, and approved the final draft.
- Carsten Paul analyzed the data, prepared figures and/or tables, authored or reviewed drafts of the paper, and approved the final draft.

### Data Availability

No raw data or code was used as this is a literature review.

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
