# Peer review of "Potential of the economic valuation of soil-based ecosystem services to inform sustainable soil management and policy"

_PeerJ, doi:10.7717/peerj.8749_

## Round 0.1 · original submission · Major Revisions

Dear authors,

Your manuscript has been reviewed by two external reviewers (one economist, one not), who found that your work has merit but also suffers from some deficiencies. Upon careful consideration, I am ready to evaluate a revised version that would take into account the comments and suggestions they have made.

In particular, please carefully address the following:
* As the paper should be of interest to non-economists also, make sure to explicit/define all the terms which might be tricky for a non-specialist to understand.
* Clarify what the "state of the art" methods actually are and what alternative approach would be recommended as a way forward (rev 1). This requires reinforcing the presentation of what has already been done in earlier publications, what the gaps are, and how the current ms. addresses these gaps (rev 2).
* Trade-offs : please provide specific examples of where economic evaluation have been/could be used to quantify trade-offs
* Both reviewers indicated that Tables 2 and 3 could be merged. information on results of each study could also be added (see suggestions from rev 2).
* The second reviewer indicated possible refinements in analysis and synthesis of the review, plus in organizing the ms. (see the suggestions made, incl. a proposed figure using drawings/pictures).
* The flow diagram could be revised by adding some examples that could help the audience to relate to.
* In section 5, be clearer on why accurately cost the contribution of soils to ecosystem services is important for.
* More generally, both reviewers have some reservation regarding the policy implications part of the paper ; try to discuss these better (for instance, recognize the limitation of the literature so far in this context; discuss the interest of qualitative versus quantitative approaches; etc.).

I look forward to receiving a revised version of your manuscript along with a point by point response to the reviewers' comments and suggestions.
best regards
Xavier

·

Basic reporting

no comment

Experimental design

no comment

Validity of the findings

no comment

Additional comments

This is a really nicely written and interesting paper on the potential for economic evaluation of soil ecosystem services, which I enjoyed reading and stimulated many questions and thoughts on the subject. As a non-specialist in the economics field, I can only offer general points for discussion and I am not qualified to offer specific technical comments on the text. My main points (and a few specific points) are listed below.

Authors state that current valuation studies are inadequate – too few ecosystem services and not using state of the art methods. These short falls are discussed in depth, but it's not clear to me what the "state of the art" methods actually are. What alternative approach would be recommended by the authors as a way forward? Could an example be given for clarity for the non-economist reader?

In a similar vein, the authors state that there is a need to express trade-offs between ecosystems services and suggest that economic valuation is the way to do this, but it is apparent from the review that most studies only focus on one or two services and cost these separately rather than take into account any trade-offs. Hedonic pricing approach is mentioned as a more advanced method for doing this, but this doesn’t seem to be defined for the non-economist anywhere and I personally don’t understand what this is (apologies!). Trade-offs are also referred to in the conclusions – line 678, but are there any specific examples of where economic evaluation have been used to quantify trade-offs?

Relatedly, given the criticisms and limitations of economic evaluation, what are the alternatives that may avoid these fundamental problems with this approach? E.g. line 270 – “valuations give an idea of “orders of magnitude” rather than precise costs. However, precise costs are then often quoted without qualification, even if they are only broad estimates. Once a precise value is attached to an object, then it becomes at risk of misinterpretation. Should these values not therefore be expressed in relative or qualitative terms (low, medium, high/ less or more?) which more accurately represent the analysis and avoid some of the dangers of misinterpretation? Are there any examples of these kinds of approaches? Also, in this same section, line 271 – what are the guidelines that are referred to? Do these exist already or do they need to be generated? If the former, could they be referred to here? If the latter, then what do they need to contain?

Authors state that the paper should be of interest to non-economists, which is really useful, but there are a few terms and phrases that are maybe tricky for a non-specialist to understand. E.g. hedonic pricing (as mentioned above), “use” and “non-use” values etc. Apologies if these are defined and I’ve just missed them, but it would help to have a brief layman’s description of these.

Section 2 – survey methodology. Just a personal preference, but the description of the survey method may be better placed at the start of section 4.2 where the lit review is reported. The overview isn't much of an overview and could do with a description of the method.

Section 4.1 – ecosystem services – here, only provisioning and regulating are mentioned? What about supporting and cultural services? … particularly as cultural services are mentioned explicitly later on (lines 285 and 332) and section 4.2.2.3

Trivial grammatical errors:
Line 171 “allows a comparison of the …”
Line 186 “we do not attempt for a …” (delete “for”)
Line 410 “disentanglement [of]…” (insert “of”)
Line 567 “…or specific public or merit good characteristics” (not sure I understand – rephrase?)
Line 589 “…a decision [of] which strategy…”(insert “of”) (and check this section (6.2) for grammar)

Table 2. I don’t think this is very useful on it’s own as just a list of references, though I can see it is necessary for the interpretation of table 3. Could these two tables be merged or combined in some way?

General comment: In section 5, there is much discussion on the factors that must be taken into account in order to more accurately cost the contribution of soils to ecosystem services, but it’s not clear what the point of this valuation is. Surely, there’s only need for a price tag if this is to be used to directly inform incentive schemes? But could incentive schemes not be based purely on how much is needed to convince a land manager to change his/her behaviour, regardless of the “actual” value of the soil/service (which, as stated previously is only a guide to the order of magnitude rather than an actual figure). Might this point be useful to bring out in the discussion? E.g. and again in section 6.2 line 611 on.

General comment: Section 6.1 – agenda setting. Line 580 “a comprehensive set of values for different ecosystem services…” Presumably these values don’t have to have a price tag associated with them? A prioritised list would be just as effective?

And another general point of discussion – section 6.3 care would need to be taken in using economic valuation of one particular component of the agroecosystem if comparable valuations are not available for other elements. So, if ecosystem services provided by soils have been costs and this is used to legitimise an incentive scheme, but the ecosystem services provided by above-ground biodiversity have not, then there is a risk that the prioritise for management interventions may be artificially skewed.

Conclusions – it would be nice to see a recommendation or description of what a more effective solution might look like. And also an opinion on whether alternatives to economic valuation (e.g. qualitative approaches) could be more or less effective (though I realise that these alternatives are outwith the specific scope of this review).
Similarly, line 681 “…needs to be complemented with other perspectives ..” Like what? Give example to make this more concrete?

·

Basic reporting

Soil is the basis of life; the physical, chemical, and biological properties of soil supports various functions individually and in combinations, which are connected to the ecological end-points and eventually to human wellbeing. The topic is of significance.

I found the article more of a book chapter than a paper. Needs refinements.

Introduction needs improvement.

Some of the materials from section 3 could go to introduction.

Perhaps adding some texts on what has already been done in earlier publications, what the gap is and how the current manuscript address the gap, would be helpful.

Experimental design

I am not sure why "Survey Methodology" is included for this manuscript.
This manuscript is based on literature review and not survey.

Refering to some more papers could also help for e.g.
Schmidt et al 2011. Persistence of soil organic matter as an ecosystem property. Nature Perspective

Brevik et al 2019 Shelter, clothing, and fuel: Often overlooked links between soils, ecosystem services, and human health. The Science of The Total Environment
Hansen and Ribaudo 2008. Economic Measures of Soil Conservation Benefits
Pereira et al Soil ecosystem services, sustainability, valuation and management.
I am skeptical about the extent of policy implications discussed with such limited studies available and reviewed.
Also is summary section needed after section 6?

Validity of the findings

Tables 2 and 3 could be merged. information on results of each study could also be added along with the year of valuation, geographic region, temporal and spatial scale, as well as whether the status or the change is estimated.

The summary can then be presented in more concise and meaningful way.

Additional comments

The topic is important and this paper is a good undertaking of the authors.I found the article more like a book chapter. Needs refinements some in analysis and synthesis of the review and second in organizing.
Soil is the basis of life; the physical, chemical, and biological properties of soil supports various functions individually and in combinations, which are connected to the ecological end-points and eventually to human wellbeing. Creating a figure along this line using drawings/pictures could be helpful.

The manuscript may also benefit from revised tables and figure. The flow diagram could be revised by adding some examples that could help the audience to relate to.
I also have reservation regarding the policy implications part of the paper.

---

## Round 0.2 · Minor Revisions

Dear authors,

Your manuscript has been much improved, which is recognized by the reviewer who has re-evaluated it. However, the reviewer recommended a final check of the ms. for correct English grammar.

During this final check, I also think that you could do a better job in analysing and presenting to what extent previous studies have analysed multiple services together. This may be done by, e;g., a network analysis based on a matrix PUBLICATIONS x SERVICES, allowing you to then build a graph (network map) presenting to what extent some services are often addressed together. In this type of representation, the size of each node (each service) depends on the number of papers assessing the service, while the links between 2 nodes relate to the number of studies addressing both services at the same time. (this is not too difficult to do, and will have a real added value)

I look forward to receiving the final version.

Best regards
Xavier LE ROUX

·

Basic reporting

Final check for correct English grammar recommended

Experimental design

no comment

Validity of the findings

no comment

Additional comments

The authors have dealt comprehensively with the reviewer recommendations from the original submission and the paper is now very much improved and I would recommend that this ms is now acceptable for publication without need for further review, once it has been thoroughly checked for English grammar.

---

## Round 0.3 · accepted · Accept

Dear authors,

I am pleased to inform you that, following your last revision of your manuscript, it is now acceptable for publication in PeerJ.

Best regards

Xavier LE ROUX